# The combination of MMSE with VSRAD and eZIS has greater accuracy for discriminating mild cognitive impairment from early Alzheimer's disease than MMSE alone

Keita Tokumitsu[1,2], Norio Yasui-Furukori[2]*, Junko Takeuchi[1], Koji Yachimori[1], Norio Sugawara[2], Yoshio Terayama[3], Nobuyuki Tanaka[3], Tatsunori Naraoka[3], Kazutaka Shimoda[2]

**1** Department of Neuropsychiatry, Towada City Hospital, Towada, Aomori, Japan, **2** Department of Psychiatry, Dokkyo Medical University School of Medicine, Mibu, Tochigi, Japan, **3** Department of Radiology, Towada City Hospital, Towada, Aomori, Japan

* furukori@dokkyomed.ac.jp

**Data Availability Statement:** The ethics committee of Towada City Hospital has set restrictions on data

## Abstract

### Background

Alzheimer's disease (AD) is assessed by carefully examining a patient's cognitive impairment. However, previous studies reported inadequate diagnostic accuracy for dementia in primary care settings. Many hospitals use the automated quantitative evaluation method known as the Voxel-based Specific Regional Analysis System for Alzheimer's Disease (VSRAD), wherein brain MRI data are used to evaluate brain morphological abnormalities associated with AD. Similarly, an automated quantitative evaluation application called the easy Z-score imaging system (eZIS), which uses brain SPECT data to detect regional cerebral blood flow decreases associated with AD, is widely used. These applications have several indicators, each of which is known to correlate with the degree of AD. However, it is not completely known whether these indicators work better when used in combination in real-world clinical practice.

### Methods

We included 112 participants with mild cognitive impairment (MCI) and 128 participants with early AD in this study. All participants underwent MRI, SPECT, and the Mini-Mental State Examination (MMSE). Demographic and clinical characteristics were assessed by univariate analysis, and logistic regression analysis with a combination of MMSE, VSRAD and eZIS indicators was performed to verify whether the diagnostic accuracy in discriminating between MCI and early AD was improved.

### Results

The area under the receiver operating characteristic curve (AUC) for the MMSE score alone was 0.835. The AUC was significantly improved to 0.870 by combining the MMSE score

sharing because the data contain potentially identifying or sensitive patient information. Please contact the institutional review board of the ethics committee of Towada City Hospital for data requests. Upon request, the ethics committee will decide whether to share the data. Contact information for our ethics committee: The institutional review board of the ethics committee of Towada City Hospital (Chairperson of the ethics committee: Dr. Masaru Kudo); Towanda City, Nishi 12-14-8, Aomori Prefecture, Japan, Postal Code 034-450 0093, Phone +81-716-23-5121, FAX +81-176-23-2999.

**Funding:** The authors received no specific funding for this work.

**Competing interests:** Norio Yasui-Furukori has been a speaker for Dainippon Sumitomo Pharmaceutical, Mochida Pharmaceutical, and MSD. Kazutaka Shimoda has received research support from Meiji Seika Pharma Co., Pfizer Inc., Dainippon Sumitomo Pharma Co., Ltd., Daiichi Sankyo Co., Otsuka Pharmaceutical Co., Ltd., Astellas Pharma Inc., Novartis Pharma K.K., Eisai Co., Ltd., Takeda Pharmaceutical Co., Ltd. and honoraria from Mitsubishi Tanabe Pharma Corporation, Meiji Seika Pharma Co., Ltd., Dainippon Sumitomo Pharma Co., Ltd., Takeda Pharmaceutical Co., Shionogi & Co., Ltd., Daiichi Sankyo Co., Pfizer Inc. and Eisai Co., Ltd. The companies had no role in the study design, the data collection or analysis, the decision to publish, or the preparation of the manuscript. The remaining authors declare that they have no competing interests to report. This does not alter our adherence to PLOS ONE policies on sharing data and materials.

with two quantitative indicators from the VSRAD and eZIS that assessed the extent of brain abnormalities.

## Conclusion

Compared with the MMSE score alone, the combination of the MMSE score with the VSRAD and eZIS indicators significantly improves the accuracy of discrimination between patients with MCI and early AD. Implementing VSRAD and eZIS does not require professional clinical experience in the treatment of dementia. Therefore, the accuracy of dementia diagnosis by physicians may easily be improved in real-world primary care settings.

## Introduction

Dementia is an important disease characterized by progressive cognitive impairment and social dysfunction [1]. In particular, Alzheimer's disease (AD) accounts for approximately 70% of dementia cases [2] and often occurs in patients in their 70s and 80s. Additionally, the prevalence increases exponentially with aging [2]. The diagnosis of dementia is assessed by carefully examining a patient's cognitive impairment and function in daily life according to international diagnostic criteria such as the International Statistical Classification of Diseases and Related Health Problems 10th edition (ICD-10) and the Diagnostic and Statistical Manual of Mental Disorders 5th edition (DSM-5) [3]. More than half of patients with mild cognitive impairment (MCI) progress to dementia within 5 years, but some MCI patients may remain MCI stable or return to normal cognition over time [4–6]. For this reason, accurate discrimination between MCI and early AD is important [4, 7, 8], especially when considering therapeutic interventions and the prognosis of dementia [4, 8, 9].

However, due to the rapid increase in the number of patients, diagnosis and treatment are not always performed by doctors who have clinical experienced with dementia. Therefore, the Mini-Mental State Examination (MMSE) is being developed as a screening tool to assist in the diagnosis of dementia [10]. This tool is a simple test consisting of questions asked by an evaluator and is often used in primary care settings. A recent meta-analysis reported that the MMSE has a sensitivity of 78.4% and a specificity of 87.8% in distinguishing between AD and MCI in primary care settings [11].

However, it is difficult to exclude cerebral organic diseases with only the MMSE, and the accuracy of this examination in distinguishing MCI from AD is inferior to the diagnostic accuracy of a specialist [11]. Therefore, it is necessary to try to improve the accuracy by combining it with other assessment methods.

At this point, brain imaging analyses are useful in the differential diagnosis of dementia and are often used in a qualitative manner to exclude organic disorders such as stroke, brain tumors, normal pressure hydrocephalus, and encephalitis. Recently, the performance of brain imaging analyses has improved, and the quantitative analysis of brain morphology and function has become possible, making it a powerful auxiliary tool for dementia diagnosis [12, 13].

It has been reported that medial temporal lobe atrophy is a characteristic morphological change in AD. The automated quantitative evaluation application called the Voxel-based Specific Regional Analysis System for Alzheimer's Disease (VSRAD), which uses brain magnetic resonance imaging (MRI) data to assess the brain morphological abnormalities associated with AD, was developed by Dr. Matsuda and colleagues [14, 15]. VSRAD applies voxel-based morphometry (VBM), which is a method for superimposing plane tomographic images from head

MRI and dividing the entire brain into small cubes for statistical analysis [16]. This free software application was updated into VSRAD advance 2 in May 2015, and it is being used in many hospitals. In particular, a Z-score of gray matter atrophy in the volume of interest (VOI) relevant to AD, which measures the severity of medial temporal atrophy, is a representative indicator of VSRAD [15, 17, 18].

Furthermore, characteristic cerebral blood flow decreases in the parietal lobe and posterior cingulate gyrus associated with AD can be assessed by single photon emission computed tomography (SPECT) [19]. An automated quantitative evaluation application called the easy Z-score imaging system (eZIS), which uses brain SPECT data to detect the regional cerebral blood flow decrease in AD, is widely used in Japan [20]. "Severity", which is one of the quantitative indicators displayed as the Z-score for regional blood flow decrease, is regarded as the most representative eZIS [21].

The VSRAD and eZIS applications have indicators, and each indicator has been reported to correlate with cognitive decline independently [12]. However, previous studies have the limitation of a small sample size, and it is not completely known whether these indicators work better in combination in real-world clinical practice. To address this clinical question, it is necessary to perform multivariate analysis. Hence, we performed a binomial logistic regression analysis combining MMSE scores and VSRAD and eZIS indicators to verify whether the diagnostic accuracy for discriminating between MCI and early AD was improved.

## Materials and methods

We recruited MCI and early AD participants from the outpatient department of Towada City Hospital between September 2016 and March 2020. All participants underwent MRI, SPECT, and a battery of laboratory tests, including assessment of thyroid function and vitamin B12, folate and serum ammonia concentrations. Cognitive function was assessed with the MMSE [22], the Revised Hasegawa's Dementia Scale (HDS-R) [23], the clock-drawing test (CDT) [24], Kohs block design test [25], and Benton visual retention test [26]. A diagnosis of AD was made based on the DSM-5 and ICD-10. A diagnosis of MCI was made according to Petersen's criteria [27]. We included patients in our study with MMSE scores of 20 or higher to exclude moderate to severe dementia [28]. The exclusion criteria were symptoms of depression, dementia with Lewy bodies, cerebrovascular disease, or any other psychiatric disorder.

### MRI procedure

MRI was performed on a 1.5T system (GE Signa Explore, General Electric Co, Boston, USA). Axial, coronal and sagittal T1-weighted sequence (SE) images (repetition time [TR], 520 ms; echo time [TE], 12.0 ms; 5-mm slice thickness) and axial T2-weighted SE images (TR, 3800 ms; TE, 97.0 ms) were obtained for diagnosis. Then, 3D volumetric acquisition of a T1-weighted gradient-echo sequence produced a gapless series of thin sagittal sections using a magnetization-prepared rapid-acquisition gradient-echo sequence (TR, 12.3 ms; TE, 5.1 ms; flip angle, 15˚; acquisition matrix, 256 × 256; 1.4-mm slice thickness).

### Voxel-based Specific Regional Analysis System for Alzheimer Disease (VSRAD)

The voxel-based analysis system in the present study has been validated [14]. Currently, the software is distributed in Japan under the name Voxel-based Specific Regional Analysis System for Alzheimer Disease advance 2 (VSRAD advance 2, Eisai Co, Tokyo, Japan). In VSRAD advance 2, the DARTEL (diffeomorphic anatomical registration through exponentiated Lie algebra) and SPM8 (Statistical Parametric Mapping 8, Institute of Neurology, London, UK) divide the T1-weighted brain MRI into cerebrospinal fluid, gray matter and white matter, and

then, anatomical standardization is performed. Normalized patient data are smoothed with a Gaussian kernel of 8-mm full width at half maximum, and the degree of brain atrophy is assessed for each voxel [29].

VSRAD scores reflect the severity of gray matter loss across the entire brain because the software compares an image with the original normal database template. VSRAD advance 2 automatically calculates the four indicators of AD shown below:

1. the Z-score of gray matter atrophy severity in the volume of interest of AD ("VSRAD VOI severity") = ((normal control average of voxel level–patient's voxel level)/normal control standard deviation),

2. the extent of gray matter atrophy in the VOI of AD ("VSRAD VOI extent") = ((number of voxels judged to have a Z-score of more than 2/number of all voxels in the volume of the hippocampus) × 100%),

3. The extent of gray matter atrophy in the whole brain ("VSRAD GM extent") = a percentage of voxels with a Z-score >2 compared with the whole brain, and

4. the ratio of the extent of gray matter atrophy in the VOI to the whole brain ("VSRAD VOI ratio") = ((number of voxels judged to have a Z-score of more than 2/number of all voxels in the volume of the whole brain) × 100%).

These four indicators of VSRAD have been explained in previous reports [15, 30].

## SPECT procedure

The patient received a bolus injection of 99mTc-ethyl cysteinate dimer (ECD) (600 MBq, Fujifilm Toyama Chemical Co, Tokyo, Japan) via the right brachial vein in a comfortable supine position with eyes closed, while awake in quiet surroundings. Twenty minutes after angiography, SPECT images were obtained using a rotating, two-head gamma camera (GE Infinia, General Electric Co, Boston, USA) with low energy high resolution and parallel hole collimators (128 × 128 matrix). The images were reconstructed using Butterworth and ramp filters, and attenuation correction was performed according to Chang's method.

## The easy Z-score imaging system (eZIS)

The eZIS calculates the degree of decrease in cerebral blood flow in each voxel after anatomical standardization of the patient's brain SPECT data by SPM2 (Statistical Parametric Mapping 2, Wellcome Department of Cognitive Neurology, London, UK).

The images are spatially normalized to an original template by using SPM2, and then, images are smoothed with a Gaussian kernel, 12 mm in full width at half maximum [21].

Subsequently, a voxel-based analysis is performed using a Z-score map calculated through a comparison of a patient's data with a control database after voxel normalization to global mean cerebral blood flow, Z-score = ([control mean] − [individual value])/(control SD).

The eZIS automatically calculates the following three indicators for characterizing regional cerebral blood flow (rCBF):

1. The severity of rCBF decrease in a specific region showing rCBF reduction from the averaged positive Z-score in the voxels of interest (bilateral posterior cingulate cortices [PCC], precunei, and parietal cortices) ("eZIS severity"),

2. The extent of a significant regional rCBF reduction in the voxel of interest by calculating the percentage of coordinates with a Z-value exceeding the threshold value of 2 ("eZIS extent"), and

3. The ratio of the extent of a region showing significant rCBF reduction in the voxel of interest to the extent of a region showing significant rCBF reduction in the whole brain, which is also the percentage of coordinates with a Z-value exceeding the threshold value of 2 ("eZIS ratio"); this ratio indicates the specificity of the rCBF reduction in the voxel of interest compared with that in the whole brain.

These three indicators of eZIS have been explained in previous reports [20, 31].

## Statistical analysis

All statistical analyses were performed with EZR (Saitama Medical Center, Jichi Medical University, Saitama, Japan) [32], which is a graphical user interface for R (The R Foundation for Statistical Computing, Vienna, Austria, version 3.5.2). More precisely, it is a modified version of the R commander that was designed to add statistical functions frequently used in biostatistics.

First, all statistical tests were based on a two-sided significance level of $p < 0.05$. Demographic and clinical characteristics were analyzed using the chi-square test and Mann–Whitney U test for differences between MCI and early AD patients. For multiple univariate analysis, the Benjamini-Hochberg procedure was used to determine whether each p-value was statistically significant.

Second, a forward-backward stepwise binomial logistic regression analysis based on Akaike's information criterion (AIC) was performed. MCI or early AD were included in the analysis as dependent variables, and sex, age, education year, MMSE score, VSRAD VOI severity, VSRAD VOI extent, VSRAD GM extent, VSRAD ratio, eZIS severity, eZIS extent and eZIS ratio were used as candidate independent variables. Factors showing significant differences in the univariate analysis were included in the model by the stepwise method. The result of this calculation was named the "stepwise selection model".

Third, receiver operating characteristic (ROC) curve and area under the ROC curve (AUC) analyses for discrimination between MCI and early AD were performed for each VSRAD indicator, each eZIS indicator, and the stepwise selection model. A diagnosis based on the DSM-5 and ICD-10 by psychiatrists certified by the Japanese Society of Psychiatry and Neurology was used as the gold standard.

## Ethics

This study was conducted in accordance with the Declaration of Helsinki and the Japanese Ethical Guidelines for Medical and Health Research Involving Human Subjects. Prior to the initiation of the study, the study protocol was reviewed and approved by the institutional review board of the ethics committee of Towada City Hospital (No. 1–4, Approved 12 June 2020). Since this was a retrospective medical record survey, informed consent was exempted, but we instead released information on this research so that patients were free to opt out.

A contact information for our ethics committee: The institutional review board of the ethics committee of Towada City Hospital (Chairperson of the ethics committee: Dr. Masaru Kudo); Towanda City, Nishi 12-14-8, Aomori Prefecture, Japan, Postal Code 034–0093, Phone +81-716-23-5121, FAX +81-176-23-2999.

## Results

### Patient characteristics and univariate analysis

A total of 411 individuals (112 with MCI and 299 with AD) were found as candidates for the participants. After excluding individuals with MMSE < 20, we included 240 participants.

**Table 1. Demographic and clinical data for participants.**

| Factor | MCI | AD | p-value | Adjusted p-value |
|---|---|---|---|---|
| Participants: N | 112 | 128 | | |
| Women: N (%) | 68 (60.7) | 89 (69.5) | 0.174 | 0.21 |
| Men: N (%) | 44 (39.3) | 39 (30.5) | | |
| Age: mean (SD) | 77.47 (6.11) | 78.28 (5.88) | 0.298 | 0.33 |
| Education year: mean (SD) | 10.88 (2.42) | 10.80 (2.62) | 0.791 | 0.791 |
| MMSE: mean (SD) | 25.93 (2.38) | 22.84 (2.05) | <0.001 | <0.05 |
| eZIS extent: mean (SD) | 11.78 (8.97) | 15.20 (10.30) | 0.007 | <0.05 |
| eZIS ratio: mean (SD) | 2.11 (1.53) | 2.47 (1.47) | 0.066 | 0.09 |
| eZIS severity: mean (SD) | 1.16 (0.31) | 1.29 (0.35) | 0.002 | <0.05 |
| VSRAD GM extent: mean (SD) | 3.69 (1.98) | 4.94 (2.74) | <0.001 | <0.05 |
| VSRAD VOI extent: mean (SD) | 14.19 (21.34) | 34.71 (30.68) | <0.001 | <0.05 |
| VSRAD ratio: mean (SD) | 3.55 (4.69) | 7.18 (6.53) | <0.001 | <0.05 |
| VSRAD VOI severity: mean (SD) | 1.16 (0.79) | 1.88 (1.07) | <0.001 | <0.05 |

Abbreviations in Table 1: mild cognitive impairment, MCI; Alzheimer's disease, AD; Voxel-based Specific Regional Analysis System for Alzheimer's Disease, VSRAD;
volume of interest, VOI; gray matter atrophy in the whole brain, GM; easy Z-score imaging system, eZIS; standard deviation, SD.
An adjusted p-value <0.05 was regarded as significant using the Benjamini-Hochberg procedure due to multiple testing.

There were 112 participants with MCI (68 women and 44 men; median age = 77.5 years) and 128 participants with early AD (89 women and 39 men; median age = 78 years).

According to a report by the Ministry of Health, Labor and Welfare, an estimated 4 million people had MCI, and 3.12 million people had AD (4.62 million with dementia) in the general population of Japan in 2012. From these data, the ratio of AD to MCI was found to be 43.8:56.2 [33]. On the other hand, in our study, the proportions of AD and MCI were 53.3% and 46.7%, respectively. The sample size for comparing the ratio of one group with the known ratio was N = 237 when calculated with a statistical power of $(1-\beta) = 0.8$ and $\alpha = 0.05$. Our study included 240 participants and was considered to meet the required sample size.

The demographic and clinical data of the participants are shown in Table 1. As a result of the univariate analysis with the Benjamini-Hochberg procedure, there were no statistically significant differences in the sex distribution, age, education year or eZIS ratio between participants with MCI and those with early AD. There were statistically significant differences in the MMSE scores, VSRAD VOI severity, VSRAD VOI extent, VSRAD GM extent, VSRAD ratio, eZIS severity and eZIS extent of the MCI and early AD groups.

## Binomial logistic regression analyses

First, a forward-backward stepwise binomial logistic regression analysis (stepwise selection model) based on AIC was performed with MCI and early AD as the dependent variables. Statistically significant factors in the univariate analysis (MMSE score, VSRAD VOI severity, VSRAD VOI extent, VSRAD GM extent, VSRAD ratio, eZIS severity and eZIS extent) were used as independent variables for stepwise binomial logistic regression analysis. As a result of this analysis, we found that a lower MMSE score (odds ratio = 0.561; $p < 0.001$), higher VSRAD VOI extent (odds ratio = 1.025; $p < 0.001$) and higher eZIS extent (OR 1.039; $p = 0.033$) were associated with early AD.

The equation for the new scores derived from the stepwise binomial logistic regression analysis for MCI and early AD screening was as follows:

Pr (case) = 1/ (1+exp (-(13.1272+0.0244*(VSRAD VOI extent) +0.0387*(eZIS extent)-0.5777*MMSE))). These results are described in Table 2.

**Table 2. Results of the binomial logistic regression analyses.**

|  | B | SE | Wald | OR (95% CI) | P-value |
|---|---|---|---|---|---|
| Stepwise selection model |  |  |  |  |  |
| Intercept | 13.127 |  |  |  |  |
| MMSE | -0.578 | 0.086 | 45.056 | 0.561 (0.474–0.664) | <0.001 |
| VSRAD VOI extent | 0.024 | 0.007 | 13.124 | 1.025 (1.011–1.038) | <0.001 |
| eZIS extent | 0.039 | 0.018 | 4.573 | 1.039 (1.003–1.077) | 0.033 |

Abbreviations in Table 2: Mini-Mental State Examination, MMSE; Voxel-based Specific Regional Analysis System for Alzheimer's Disease, VSRAD; volume of interest, VOI; easy Z-score imaging system, eZIS; regression coefficient, B; standard error, SE; odds ratio, OR; confidence interval, CI.

### Receiver operating characteristic (ROC) curve analysis

Table 3 shows the results of the ROC curve analysis for the discrimination between MCI and early AD. The AUC using MMSE scores alone was 0.835. On the other hand, the AUC obtained from the stepwise selection model that combined MMSE, VSRAD VOI extent and eZIS extent was 0.870. A chi-square test of these AUCs revealed that the stepwise selection model had a statistically significantly larger area than the MMSE score alone (p = 0.012). The results of the ROC analysis are described in Table 3 and Fig 1.

The results of the ROC analysis for the discrimination between MCI and early AD. The area under the ROC curve (AUC) using MMSE scores alone was 0.835. On the other hand, the AUC obtained from the stepwise selection model that combined MMSE, VSRAD VOI extent and eZIS extent was 0.870. A chi-square test of these AUCs revealed that the stepwise selection model had a statistically significantly larger area than the MMSE score alone (p = 0.012).)

### Discussion

Our study revealed that the diagnostic accuracy that distinguished MCI from early AD was statistically significantly improved by combining quantitative data from psychological tests with brain morphological and functional image analyses. The AUC with the MMSE scores alone was 0.835, but the AUC was improved to 0.870 by adding the VSRAD VOI extent and eZIS extent to the MMSE scores. VSRAD and eZIS are useful applications that automatically quantify cerebral atrophy and blood flow decreases based on the data obtained from MRI and

**Table 3. Results of the receiver operating characteristic curve analyses.**

|  | Cutoff point | FPF | TPF | AUC (95% CI) | SE |
|---|---|---|---|---|---|
| MMSE | 23 | 0.143 | 0.695 | 0.835 (0.784–0.886) | 0.026 |
| VSRAD VOI severity | 1.35 | 0.250 | 0.625 | 0.710 (0.645–0.775) | 0.033 |
| VSRAD VOI extent | 33.54 | 0.125 | 0.492 | 0.708 (0.643–0.773) | 0.033 |
| VSRAD GM extent | 3.51 | 0.402 | 0.703 | 0.649 (0.579–0.719) | 0.036 |
| VSRAD ratio | 5.66 | 0.223 | 0.531 | 0.677 (0.610–0.745) | 0.034 |
| eZIS severity | 1.3 | 0.250 | 0.469 | 0.616 (0.544–0.687) | 0.036 |
| eZIS extent | 13.8 | 0.277 | 0.500 | 0.607 (0.536–0.679) | 0.037 |
| eZIS ratio | 1.8 | 0.438 | 0.617 | 0.581 (0.580–0.654) | 0.037 |
| Stepwise selection model | 0.517 | 0.179 | 0.828 | 0.870 (0.824–0.916) | 0.023 |

Abbreviations in Table 3: Mini-Mental State Examination, MMSE; Voxel-based Specific Regional Analysis System for Alzheimer's Disease, VSRAD; volume of interest, VOI; gray matter atrophy in the whole brain, GM; easy Z-score imaging system, eZIS; regression coefficient, false positive fraction, FPF; true positive fraction, TPF; area under the curve, AUC; standard error, SE.

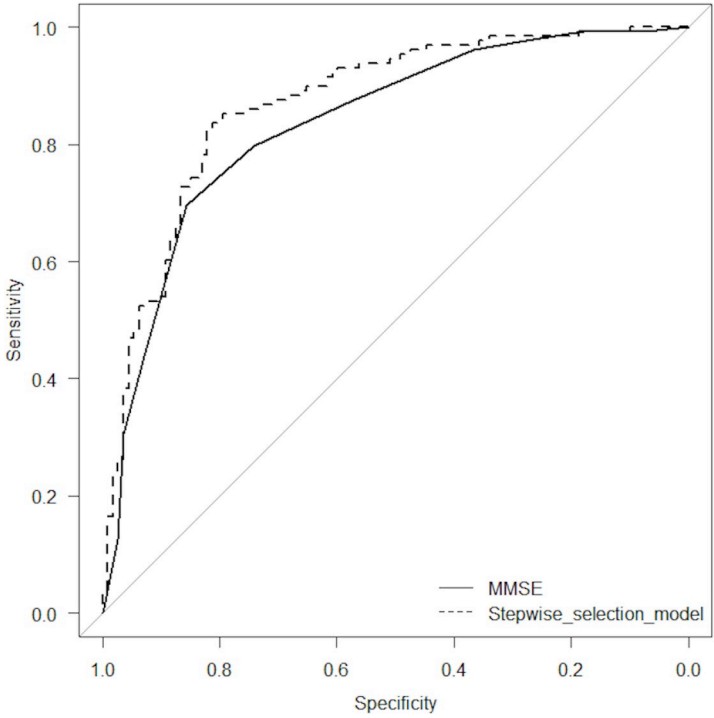

**Fig 1. Receiver operating characteristic (ROC) curve analyses.**

SPECT, so physicians may improve the diagnostic accuracy of dementia regardless of clinical experience.

For these applications, "VSRAD VOI severity" and "eZIS severity" are the most representative indicators. However, interestingly, the factors selected in the stepwise selection model were "VSRAD VOI extent" and "eZIS extent".

According to Braak staging [34, 35], which explains the pathological changes in AD, the burden of tau protein spreads as the stage progresses. In addition, gray matter loss in the medial temporal lobe has already been recognized in MCI, and it is known that the loss area expands at conversion to AD [36]. Mizumura *et al* stated that studying the "extent" of the region of abnormal blood flow that causes functional disorder is more rational than assessing the "severity" of the blood flow abnormality that reflects local tissue degeneration [37]. Their discussion is confirmed by the results of our study. Therefore, on brain MRI and SPECT, the "extent" of the lesion may be more important for distinguishing MCI from AD than the "severity" of local atrophy and decreased regional cerebral blood flow.

In our study, we compared participants with MCI and those with early AD, but VSRAD and eZIS had lower AUCs for each indicator than previous studies comparing healthy volunteers with early AD [20, 38]. MCI may have some findings that are similar to those of early AD, so there may have been relatively poor discrimination accuracy for discriminating MCI from early AD.

There are several assessment tools for dementia, but it is not fully understood which combinations work better. A previous study on positron emission tomography (PET) and MRI stated that it was important to combine modalities to assess AD from different perspectives [39]. Another study reported that the diagnostic accuracy of dementia was improved by combining two different neuropsychological tests: the MMSE and the clock-drawing test [40]. Here, we have shown that the combination of a neuropsychological test with a brain imaging

evaluation, based on logistic regression analysis, improves the diagnostic accuracy of discriminating MCI from early AD in a statistically significant manner compared with the use of each test alone.

When predicting diagnostic accuracy by combining multiple different indicators, it is important to select statistically significant indicators and weight them according to the multivariate analysis results. In our study, we included a sufficient number of patients, which was more than 10 times the number of independent variables [41]. For this reason, we could identify statistically significant independent variables not only by univariate correlation analysis but also by binomial logistic regression analysis.

Previous studies have reported inadequate diagnostic accuracy for dementia in primary care settings [42]. That is, the diagnosis of early AD may be delayed and may lead to underestimation of cognitive impairment [42]. In Japan, the number of dementia patients will exceed 6 million in 2020 due to the rapid aging of the population [43]. In addition, previous research estimated that the social cost of dementia in Japan will have reached approximately 14.5 trillion yen per year in 2014 [44]. It is known that early diagnosis of dementia and appropriate intervention not only improve the quality of life of patients and their families but also reduce socioeconomic costs [45]. Therefore, we also considered it important from the viewpoint of public health to combine psychological tests and quantitative brain imaging data to improve the accuracy and reproducibility of dementia diagnoses.

As imaging modalities evolve and examination costs decrease, it is expected that the number of diagnostic support tools will increase. Among brain imaging assessment tools, PET is a useful biomarker as are MRI and SPECT [39]; however, the use of PET for the detection of dementia has not yet been accepted for reimbursement in the National Health Insurance system in Japan. Hence, MRI and SPECT are widely applied to patients with cognitive impairment in Japan [46]. New biomarkers for the diagnosis of AD, including the measurement of cerebrospinal fluid β-amyloid 42 and tau proteins [47], are being clinically applied. It is necessary to continue conducting research on better test combinations that take cost performance and insurance adaptation into account.

Our study has several limitations. Our research was a single-center, retrospective, cross-sectional study. It can explain the diagnostic accuracy of the test, but the causal relationship between the results and the disease remains unknown. In addition, our study did not randomize the patient population, which may lead to sampling bias. In our study, the sample size was appropriate, and the statistical power was also sufficient. Although the odds ratio of the independent variable in logistic regression analysis was statistically significant, the effect size was limited. Furthermore, although the site and extent of atrophy differ based on the subtypes of AD [48], heterogeneity in the AD population may have been high because our study did not identify these subtypes. Importantly, in 2011, the National Institute on Aging and Alzheimer's Association created separate diagnostic recommendations for the preclinical, MCI, and dementia stages of AD (the NIA-AA guidelines), and the guidelines were updated in 2018 [49]. The research framework focuses on the diagnosis of AD with biomarkers grouped into those of β amyloid deposition, pathologic tau, and neurodegeneration in living persons [49]. We did not evaluate these biomarkers of AD in our study. As a result, it was not possible to accurately assess cognitive impairment and pathological abnormalities in AD based on the NIA-AA guidelines, which is a novel definition of AD continuum staging using biomarkers [49].

It is also possible that the population of MCI subjects may contain not only MCI due to AD but also MCI due to another cognitive impairment (e.g., dementia with Lewy body or frontotemporal dementia). Our research has limitations in terms of population heterogeneity. In addition, there are other psychological tests used to evaluate cognitive function in addition to

the MMSE; however, we did not include them in the statistical analysis because they had some missing values, and the listwise method did not provide a sufficient sample size. Further research on the combination of other psychological and imaging tests is needed.

## Conclusions

We found that combining the MMSE score with two indicators from automated quantitative assessment applications using brain MRI and SPECT, known as VSRAD and eZIS, respectively, significantly improved the accuracy of discrimination between MCI and early AD compared with the MMSE score alone. Implementing VSRAD and eZIS does not require professional clinical experience in the treatment of dementia. Therefore, the accuracy of dementia diagnosis by physicians may be easily improved in real-world primary care settings.

## Acknowledgments

We gratefully acknowledge Mr. Terue Urushihata for his work with psychological testing. We would like to thank all medical staff of Towada City Hospital for their kind support.

## Author Contributions

**Conceptualization:** Keita Tokumitsu.

**Data curation:** Keita Tokumitsu.

**Methodology:** Norio Sugawara.

**Project administration:** Keita Tokumitsu.

**Writing – original draft:** Keita Tokumitsu.

**Writing – review & editing:** Norio Yasui-Furukori, Junko Takeuchi, Koji Yachimori, Norio Sugawara, Yoshio Terayama, Nobuyuki Tanaka, Tatsunori Naraoka, Kazutaka Shimoda.

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
