## [Decision Letter · Decision Letter 0]

19 Nov 2020

PONE-D-20-30261

Combining MMSE and brain MRI and SPECT indicators from the automated quantitative assessment applications VSRAD and eZIS improves accuracy of discrimination between mild cognitive impairment and early Alzheimer's disease.

PLOS ONE

Dear Dr. Yasui-Furukori,

Thank you for submitting your manuscript to PLOS ONE. After careful consideration by 2 Reviewers and an Academic Editor, all of the critiques of both Reviewers must be addressed in detail in a revision to determine publication status. If you are prepared to undertake the work required, I would be pleased to reconsider my decision, but revision of the original submission without directly addressing the critiques of the 2 Reviewers does not guarantee acceptance for publication in PLOS ONE. If the authors do not feel that the queries can be addressed, please consider submitting to another publication medium. A revised submission will be sent out for re-review. The authors are urged to have the manuscript given a hard copyedit for syntax and grammar.

**Comments to the Author**

1. Is the manuscript technically sound, and do the data support the conclusions?

Reviewer #1: Yes

Reviewer #2: Yes

2. Has the statistical analysis been performed appropriately and rigorously? 

Reviewer #1: No

Reviewer #2: Yes

3. Have the authors made all data underlying the findings in their manuscript fully available?

Reviewer #1: Yes

Reviewer #2: Yes

4. Is the manuscript presented in an intelligible fashion and written in standard English?

Reviewer #1: Yes

Reviewer #2: Yes

5. Review Comments to the Author

Reviewer #1: Commentary on: Combining MMSE and brain MRI and SPECT indicators from the automated quantitative assessment applications VSRAD and [the] eZIS improve [the] accuracy of discrimination between mild cognitive impairment and early Alzheimer's disease.

The authors present a retrospective, single-center, cross-sectional study that evaluates 240 MCI and early AD patients with MMSE, voxel-base morphometry (VBM), and cerebral blood flow (CBF) quantification with SPECT. VBM medial temporal atrophy and decreased CBF in the parietal lobe and posterior cingulate gyrus are used as proxies of an AD profile. The authors claim that by combining two proxies for neuronal injury (i.e. decreased brain volume and decreased CBF) and a single neuro-psychometric assessment score clinicians can acceptably differentiate between AD and MCI.

Major remarks

Only one screening tool is employed for cognitive assessment. Please explain why other and complementary neuropsychological screening tools were not used (e.g. MoCA, CDR-SB, Mattis Dementia Rating Scale).

Please briefly summarize image preprocessing steps for both VBM and SPECT imaging.

Please provide the logistic regression equations. Please state how multiple comparisons were corrected for.

For the ROC and AUC analyses, it is not clear what was used as the gold standard to classify possible AD. I would refer the authors to the work by the NIA-AA 2018 research framework to understand what the novel definitions of AD continuum staging are.

In the introduction, the authors mention that “previous studies have the limitation that the sample size is small, and it is not completely known whether these indicators work better in combination in real-world clinical practice”; the reader would expect that the authors would provide a sample calculation for each outcome measure (i.e. MMSE, VMB, and CBF) to differentiate between MCI and early AD. Plenty of literature exists for the use of MMSE; however, although there is plenty of data regarding VBM and CBF to distinguish between MCI and AD, the reader might question how adequate are the VBM and CBF AD profile proxies that the authors used.

It is not clear what are the advantages of using the forced entry model over only using the stepwise logistic regression model. The authors claim that the advantage of the forced entry model is, “the odds ratio for the indicators in the forced entry model was higher than that on the stepwise selection model. Therefore, it was suggested that the statistical impact and clinical influence of each indicator may be different”. These are not valid reasons to select a different statistical logistic regression model.

Minor remarks

Please consider using a brief and descriptive title.

Please revise and correct the manuscript for content structure.

In the introduction the authors mention the diagnostic criteria based on ICD-10 and DSM-5, which is fine; however, the NIA-AA guidelines and definitions are most often used in our field. Please review these guidelines to discuss some of the limitations that this study has based on the disease definitions used.

The authors mention that “More than half of patients with mild cognitive impairment (MCI) progress to dementia within 5 years, but some MCI patients may remain MCI stable or return to normal cognition over time [4]”. The topic of AD progression, reversion, and cognitive stable MCI patients varies depending on the type of cognitive assessment and mediating factors such as cognitive and brain reserve, which is beyond the scope of this study; however, if the authors want to present this information, more than one reference regarding the rates of progression is necessary.

Please include a review of the literature about the diagnostic accuracy of the MMSE for dementia and MCI (e.g. Cochrane Reviews).

Please clarify what the authors mean by “The severity of regional blood flow decrease is the most representative quantitative indicator in eZIS”.

What is meant by “These two applications have several indicators, each of which is known to correlate with the degree of AD [10]”?

Please include more detail about the measures of dispersion among the group descriptions in the results section.

Please provide 95% confidence intervals in the tables and in the text.

Please place the tables at the end of the manuscript.

The authors claim in the discussion section that the participants were assessed for social and emotional function, “Since cognitive function was evaluated not only for memory but also for social and emotional function, we thought patients with relatively widespread neuronal loss and decreased cerebral blood flow were more likely to progress to dementia”; however, no evidence of this is found in the text.

Final remarks

Please have a language editor revise the manuscript. The research question lacks novelty. The authors fail to provide sufficient evidence that the study has sufficient statistical power. The authors use normalized VMB and CBF measures which are technically sound but questionably underpowered. The authors only use one proxy for cognitive impairment, which is a major weakness of the study design. The effect sizes in the multivariate analysis are small (i.e. OR of 0.561, 1.025, 1.039; 0.567, 1.999, 2.994) and no correction for multiple comparisons is applied, rendering some of the differences not significant.

Reviewer #2: The authors aimed to demonstrate that combining MMSE score with two automated quantitative methods VSRAD (based on brain MRI data) and eZIS (based on brain SPECT data) in differentiate between patients with MCI and early AD has a greater accuracy than MME score alone.

The topic is very interesting and it could have a clinical impact.

The paper is specific, well-structured and precise in the technical description.

The title is inherent in the purpose of the study, but too long and may not catch the reader's attention.

The aim of the study is clearly exposed and well-argued.

The methods and the statistic evaluation are described in an accurate way.

The results are complete, presented in a logically corrected sequence and they are analyzed extensively in the discussion. Moreover, the authors underline the applicative value of the results in real-world.

Tables and graphics show adequate drafting modalities, sufficient length, layout and size.

Bibliographic references are appropriate and consistent.

The manuscript needs the following revisions:

• Title: it is preferable to review it, reducing its length and making it more captivating.

• Results: (Patient characteristics) the lines 248-256 could be reduced to a single introductory sentence of Table 1, which is already explanatory and complete with all data.

6. PLOS authors have the option to publish the peer review history of their article (what does this mean?). If published, this will include your full peer review and any attached files.

**Do you want your identity to be public for this peer review?** For information about this choice, including consent withdrawal, please see our Privacy Policy.

Reviewer #1: **Yes: **Jaime Daniel Mondragon

Reviewer #2: No

We look forward to receiving your revised manuscript.

Kind regards,

Stephen D. Ginsberg, Ph.D.

Section Editor

PLOS ONE

"Norio Yasui-Furukori has been a speaker for Dainippon-Sumitomo Pharmaceutical, Mochida Pharmaceutical, and MSD. Kazutaka Shimoda has received research support from Meiji Seika Pharma Co., Pfizer Inc., Dainippon Sumitomo Pharma Co., Ltd., Daiichi Sankyo Co., Otsuka Pharmaceutical Co., Ltd., Astellas Pharma Inc., Novartis Pharma K.K., Eisai Co., Ltd., Takeda Pharmaceutical Co., Ltd. and honoraria from Mitsubishi Tanabe Pharma Corporation, Meiji Seika Pharma Co., Ltd., Dainippon Sumitomo Pharma Co., Ltd., Takeda Pharmaceutical Co., Shionogi & Co., Ltd., Daiichi Sankyo Co., Pfizer Inc. and Eisai Co., Ltd. The companies had no role in the study design, the data collection and analysis, the decision to publish, or the preparation of the manuscript. The remaining authors declare that they have no competing interests to report.".

i) Please confirm that this does not alter your adherence to all PLOS ONE policies on sharing data and materials, by including the following statement: "This does not alter our adherence to  PLOS ONE policies on sharing data and materials.” (as detailed online in our guide for authors http://journals.plos.org/plosone/s/competing-interests). If there are restrictions on sharing of data and/or materials, please state these. Please note that we cannot proceed with consideration of your article until this information has been declared.

ii) Please include your updated Competing Interests statement in your cover letter; we will change the online submission form on your behalf.

---

## [Author Response · Author response to Decision Letter 0]

17 Jan 2021

Author’s response to the reviewers:

We are grateful to the reviewers for their critical comments and useful suggestions, which have helped us improve our paper. As indicated in the responses that follow, we have taken all these comments and suggestions into account in the revised version of our paper.

We hope that the revised version of our paper is now suitable for publication in PLOS ONE.

Sincerely,

Norio Yasui-Furukori, MD, PhD

PONE-D-20-30261

“The combination of MMSE with VSRAD and eZIS has greater accuracy for discriminating mild cognitive impairment from early Alzheimer’s disease than MMSE alone.” (We revised title based on review comments)

Reviewer #1:

Comment:

Major remarks

Only one screening tool is employed for cognitive assessment. Please explain why other and complementary neuropsychological screening tools were not used (e.g. MoCA, CDR-SB, Mattis Dementia Rating Scale).

Response:

Thank you for your important observation. Regarding the screening tool for cognitive function, in addition to the MMSE, other tests including the HDS-R, CDT, Kohs block design test, and Benton visual retention test were carried out by clinical psychologists. This missing information was added, and the manuscript was revised.

(Revised Manuscript with Track Changes: Lines 135-138.)

Comment:

Please briefly summarize image preprocessing steps for both VBM and SPECT imaging.

Response:

 In VSRAD advance 2, the DARTEL (diffeomorphic anatomical registration through exponentiated Lie algebra) and SPM8 (Statistical Parametric Mapping 8, Institute of Neurology, London, UK) divide the T1-weighted brain MRI into cerebrospinal fluid, gray matter and white matter, and then, anatomical standardization is performed. Normalized patient data are smoothed with a Gaussian kernel of 8-mm full width at half maximum, and the degree of brain atrophy is assessed for each voxel.

 On the other hand, the eZIS calculates the degree of decrease in cerebral blood flow in each voxel after anatomical standardization of the patient's brain SPECT data by SPM2 (Statistical Parametric Mapping 2, Wellcome Department of Cognitive Neurology, London, UK).

The images are spatially normalized by using SPM2 to an original template, then images are smoothed with a Gaussian kernel, 12 mm in full width at half maximum, and a voxel-based analysis is performed.

 We revised the manuscript according to the reviewer’s comments.

(Revised Manuscript with Track Changes: Lines 160-166, 195-200.)

Comment:

Please provide the logistic regression equations. Please state how multiple comparisons were corrected for.

Response:

The equation for the new scores derived from the stepwise binomial logistic regression analysis for MCI and early AD screening was as follows:

Pr (case) = 1/ (1+exp (-(13.1272+0.0244*(VSRAD VOI extent) +0.0387*(eZIS extent)-0.5777*MMSE))). In the multiple univariate analyses shown in Table 1, the Benjamini-Hochberg procedure was used to correct the multiple comparisons. We added adjusted p-value in Table 1. On the other hand, the data obtained from logistic regression analysis did not need to be modified because the logit model itself shows the corrected odds ratio. However, repeating the tests with different logit models increases the likelihood of type I errors. Based on the reviewer’s comment, the analysis result with the forced entry model was deleted, so we thought that it is not necessary to revise the result of the binomial logistic regression analysis by the stepwise model again.

(Revised Manuscript with Track Changes: Lines 229-231,311-312, 324-327 and Table 1.)

Comment:

For the ROC and AUC analyses, it is not clear what was used as the gold standard to classify possible AD. I would refer the authors to the work by the NIA-AA 2018 research framework to understand what the novel definitions of AD continuum staging are.

Response:

Thank you for your suggestions. In our research, a diagnosis based on the DSM-5 and ICD-10 by psychiatrists certified by the Japanese Society of Psychiatry and Neurology was used as the gold standard. On the other hand, we referred to the NIA-AA 2018 research framework and described the limitations of our research.

(Revised Manuscript with Track Changes: Lines 247-249, 457-469.)

Comment:

In the introduction, the authors mention that “previous studies have the limitation that the sample size is small, and it is not completely known whether these indicators work better in combination in real-world clinical practice”; the reader would expect that the authors would provide a sample calculation for each outcome measure (i.e. MMSE, VMB, and CBF) to differentiate between MCI and early AD. Plenty of literature exists for the use of MMSE; however, although there is plenty of data regarding VBM and CBF to distinguish between MCI and AD, the reader might question how adequate are the VBM and CBF AD profile proxies that the authors used.

Response:

We appreciate your insightful comments.

According to a report by the Ministry of Health, Labor and Welfare, an estimated 4 million people had MCI, and 3.12 million people had Alzheimer's disease (4.62 million with dementia) in the general population of Japan in 2012. From these data, the ratio of AD to MCI was found to be 43.8:56.2. (https://www.mhlw.go.jp/content/12300000/000519620.pdf).

On the other hand, in our study, the proportions of AD and MCI were 53.3% and 46.7%, respectively. The sample size for comparing the ratio of one group with the known ratio was N = 237 when calculated with a statistical power of (1-β) = 0.8 and α = 0.05. Our study included 240 participants and was considered to meet the required sample size. If there is a statistically significant difference (adjusted p-value <0.05) for each result measurement using these samples, it can be determined that these indicators can distinguish between MCI and AD.

(Revised Manuscript with Track Changes: Lines 273-280.)

Comment:

It is not clear what are the advantages of using the forced entry model over only using the stepwise logistic regression model. The authors claim that the advantage of the forced entry model is, “the odds ratio for the indicators in the forced entry model was higher than that on the stepwise selection model. Therefore, it was suggested that the statistical impact and clinical influence of each indicator may be different”. These are not valid reasons to select a different statistical logistic regression model.

Response:

Thank you for your valuable feedback. As the reviewer noted, due to the lack of a valid reason for using the forced entry model in addition to the stepwise selection model, the results of the forced entry model for the binomial logistic regression analysis have been removed. We considered the reviewer's comments to be reasonable, as repeating tests with multiple models is more likely to cause type I errors.

(Revised Manuscript with Track Changes: Lines 252-230, 347-348, 350-353, 401-408. Table 2 and 3.)

Comment:

Minor remarks

Please consider using a brief and descriptive title.

Response:

According to the reviewer’s suggestion, we have shortened the title. The previous title “Combining MMSE and brain MRI and SPECT indicators from the automated quantitative assessment applications VSRAD and eZIS improves accuracy of discrimination between mild cognitive impairment and early Alzheimer's disease.” changed to "The combination of MMSE with VSRAD and eZIS has greater accuracy for discriminating mild cognitive impairment from early Alzheimer’s disease than MMSE alone".

(Revised Manuscript with Track Changes: Lines 3-5, 7-9.)

Comment:

Please revise and correct the manuscript for content structure.

Response:

The structure of the manuscript has been revised. In particular, we have increased the number of references for changes in cognitive function and removed speculative statements and results for the forced entry model for binomial logistic regression analysis. In addition, the manuscript was proofread for English language by American Journal Experts. If necessary, we will submit an editing certificate (https://www.aje.com/).

Comment:

In the introduction the authors mention the diagnostic criteria based on ICD-10 and DSM-5, which is fine; however, the NIA-AA guidelines and definitions are most often used in our field. Please review these guidelines to discuss some of the limitations that this study has based on the disease definitions used.

Response:

Thank you for introducing us to the NIA-AA guidelines. In 2011, 

the National Institute on Aging and Alzheimer's Association created separate diagnostic recommendations for the preclinical, MCI, and dementia stages of AD (the NIA-AA guidelines), and the guidelines were updated in 2018. The research framework focuses on the diagnosis of AD with biomarkers grouped into those of β amyloid deposition, pathologic tau, and neurodegeneration in living persons. We did not evaluate these biomarkers of AD in our study. As a result, it was not possible to accurately assess cognitive impairment and pathological abnormalities in AD based on the NIA-AA guidelines, which is a novel definition of AD continuum staging using biomarkers.

 It is also possible that the population of MCI subjects may contain not only MCI due to AD but also MCI due to another cognitive impairment (e.g., dementia with Lewy body or frontotemporal dementia).

The authors of the NIA-AA guidelines emphasized that “it is premature and inappropriate to use this research framework in general medical practice” and that “this research framework should not be used to restrict alternative approaches to hypothesis testing that do not use biomarkers”. However, we thought it is important to focus on the biological perspectives of these guidelines because our study has limitations based on population heterogeneity.

(Revised Manuscript with Track Changes: Lines 457-469.)

Comment:

The authors mention that “More than half of patients with mild cognitive impairment (MCI) progress to dementia within 5 years, but some MCI patients may remain MCI stable or return to normal cognition over time [4]”. The topic of AD progression, reversion, and cognitive stable MCI patients varies depending on the type of cognitive assessment and mediating factors such as cognitive and brain reserve, which is beyond the scope of this study; however, if the authors want to present this information, more than one reference regarding the rates of progression is necessary.

Response:

The accuracy of distinguishing between MCI and Alzheimer's disease has great influence on the explanation of prognosis and the treatment content. In particular, information about the rate of progression is essential in shared decision-making with patients. For this reason, we added the reference in the introduction section.

(Revised Manuscript with Track Changes: Line 83.)

Comment:

Please include a review of the literature about the diagnostic accuracy of the MMSE for dementia and MCI (e.g. Cochrane Reviews).

Response:

Regarding the accuracy of discrimination between AD and MCI using the MMSE in the primary care setting, the sensitivity is 78.4%, and the specificity is 87.8% according to a recent meta-analysis reported by AJ Mitchell et al. 

Based on the reviewer’s comment, we added a review of the literature in the introduction section.

(Revised Manuscript with Track Changes: Lines 90-92.)

Comment:

Please clarify what the authors mean by “The severity of regional blood flow decrease is the most representative quantitative indicator in eZIS”.

Response:

Thank you for your valuable feedback. The eZIS compares the subject's cerebral blood flow data with preconfigured standard data for each voxel. Next, the eZIS program calculates the degree of cerebral blood flow hypoperfusion as the standard deviation. The absolute value of this standard deviation is defined as the "severity" of an eZIS indicator and displayed as the Z-score. The Z-score is regarded as the most representative quantitative indicator by the eZIS creator. (Kanetaka H, et al. Effects of partial volume correction on discrimination between very early Alzheimer's dementia and controls using brain perfusion SPECT. European journal of nuclear medicine and molecular imaging. 2004;31(7):975-80.)

(Revised Manuscript with Track Changes: Lines 116-122.)

Comment:

What is meant by “These two applications have several indicators, each of which is known to correlate with the degree of AD [10]”?

Response:

The VSRAD and eZIS applications have indicators including “severity”, “extent”, and “ratio”, each of which has been independently reported to correlate with cognitive decline. We have revised the sentences in our manuscript to make this easier to understand.

(Revised Manuscript with Track Changes: Lines 120-122.)

Comment:

Please include more detail about the measures of dispersion among the group descriptions in the results section.

Response:

We present the mean and SD (standard deviation) for each item as dispersion measurements in Table 1 of the Results section.

(Revised Manuscript with Track Changes: Line 310. Table 1.)

Comment:

Please provide 95% confidence intervals in the tables and in the text.

Response:

We added 95% confidence intervals for ORs in Table 2 and AUCs in Table 3.

(Revised Manuscript with Track Changes: Lines 339, 373, Table 2 and 3.)

Comment:

Please place the tables at the end of the manuscript.

Response:

Thank you for your suggestion.

PLOS ONE's submission guidelines state that "Tables should be included directly after the paragraph in which they are first cited". Therefore, please allow us to follow the journal requirements for table locations. (https://journals.plos.org/plosone/s/file?id=wjVg/PLOSOne_formatting_sample_main_body.pdf)

Comment:

The authors claim in the discussion section that the participants were assessed for social and emotional function, “Since cognitive function was evaluated not only for memory but also for social and emotional function, we thought patients with relatively widespread neuronal loss and decreased cerebral blood flow were more likely to progress to dementia”; however, no evidence of this is found in the text.

Response:

Based on the reviewer comments, we deleted sentences that were too speculative.

(Revised Manuscript with Track Changes: Lines 398-400.)

Comment:

Final remarks

Please have a language editor revise the manuscript. The research question lacks novelty. The authors fail to provide sufficient evidence that the study has sufficient statistical power. The authors use normalized VMB and CBF measures which are technically sound but questionably underpowered. The authors only use one proxy for cognitive impairment, which is a major weakness of the study design. The effect sizes in the multivariate analysis are small (i.e., OR of 0.561, 1.025, 1.039; 0.567, 1.999, 2.994) and no correction for multiple comparisons is applied, rendering some of the differences not significant.

Response:

Thank you for your suggestion.

We asked American Journal Experts to proofread the text for English and revised the manuscript. If necessary, we will submit an editing certificate (https://www.aje.com/).

In our study, the sample size was appropriate, and the statistical power was sufficient. Although the odds ratio of the independent variable in the logistic regression analysis was statistically significant, the effect size was limited.

(Revised Manuscript with Track Changes: Lines 273-280, 452-455.)

Previous researchers reported that both indicators obtained by VSRAD and eZIS are statistically significantly correlated with the severity of cognitive impairment. For this reason, VSRAD and eZIS were effective as testing methods and auxiliary diagnostic tools for cognitive impairment. In addition, our study analyzed the most effective combinations of these indicators.

(Revised Manuscript with Track Changes: Lines120-124.)

In the past, there have been no reports indicating that the combination of specific indicators for VSRAD and eZIS with the MMSE improves diagnostic accuracy over the evaluation of the MMSE alone. For this reason, we believe that our research has a novelty.

The methods section was revised because, in fact, the patients were screened by a clinical psychologist with a test battery that combined multiple neuropsychological tests. However, neuropsychological tests other than the MMSE were not included in the statistical analysis because they had some missing values, and the listwise method did not provide a sufficient sample size. Further research is needed on the combination of neuropsychological tests other than the MMSE and eZIS and VSRAD. We have described this point as a limitation of our research. 

(Revised Manuscript with Track Changes: Lines 135-138, 469-473.)

In multiple univariate analysis, the Benjamini-Hochberg procedure was used to correct multiple comparisons. On the other hand, the data obtained from logistic regression analysis did not need to be modified because the logit model itself shows the corrected odds ratio.

(Revised Manuscript with Track Changes: Lines 229-231.)

However, repeating the tests with different logit models increases the likelihood of type I errors. Based on the reviewer’s comment, the result of the analysis with the forced entry model was deleted, so we thought that it is not necessary to revise the result of the multivariate analysis with the stepwise model again.

(Revised Manuscript with Track Changes: Lines 239-243, 347-348, 350-353, 401-408. Table 2 and 3.)

Reviewer #2:

Comment:

The topic is very interesting and could have a clinical impact.

The paper is specific, well-structured and precise in the technical description.

Response:

Thank you for your valuable feedback.

Comment:

The title is inherent in the purpose of the study, but too long and may not catch the reader's attention.

Response:

We have shortened the title and revised it to be more captivating. The previous title “Combining MMSE and brain MRI and SPECT indicators from the automated quantitative assessment applications VSRAD and eZIS improves accuracy of discrimination between mild cognitive impairment and early Alzheimer's disease” was changed to "The combination of MMSE with VSRAD and eZIS has greater accuracy for discriminating mild cognitive impairment from early Alzheimer’s disease than MMSE alone".

(Revised Manuscript with Track Changes: Lines 3-5, 7-9)

Comment:

The aim of the study is clearly exposed and well-argued.

The methods and the statistic evaluation are described in an accurate way.

The results are complete, presented in a logically corrected sequence and they are analyzed extensively in the discussion. Moreover, the authors underline the applicative value of the results in real-world.

Tables and graphics show adequate drafting modalities, sufficient length, layout and size.

Bibliographic references are appropriate and consistent.

Response:

Thank you for your encouraging remarks.

Comment:

The manuscript needs the following revisions:

• Title: it is preferable to review it, reducing its length and making it more captivating.

Response:

We have shortened the title and fixed it more captivating. The previous title “Combining MMSE and brain MRI and SPECT indicators from the automated quantitative assessment applications VSRAD and eZIS improves accuracy of discrimination between mild cognitive impairment and early Alzheimer's disease” was changed to "The combination of MMSE with VSRAD and eZIS has greater accuracy for discriminating mild cognitive impairment from early Alzheimer’s disease than MMSE alone".

(Revised Manuscript with Track Changes: Lines 3-5, 7-9.)

Comment:

• Results: (Patient characteristics) the lines 248-256 could be reduced to a single introductory sentence of Table 1, which is already explanatory and complete with all data.

Response:

We summarized the description of the results in Table 1 by removing unnecessary sentences.

(Revised Manuscript with Track Changes: Lines 282-289 and Table 1.)

Comment:

Response:

We revised the manuscript following PLOS ONE’s style requirements, particularly the file name and template of the title page.

Comment:

2. PLOS requires an ORCID iD for the corresponding author in Editorial Manager on papers submitted after December 6th, 2016. Please ensure that you have an ORCID iD and that it is validated in Editorial Manager. To do this, go to ‘Update my Information’ (in the upper left-hand corner of the main menu) and click on the Fetch/Validate link next to the ORCID field. This will take you to the ORCID site and allow you to create a new iD or authenticate a pre-existing iD in Editorial Manager. Please see the following video for instructions on linking an ORCID iD to your Editorial Manager account: https://www.youtube.com/watch?v=_xcclfuvtxQ

Response:

The ORCID id of the Corresponding Author (Dr. Norio Yasui-Furukori) has been registered in the Editorial Manager account.

"Norio Yasui-Furukori has been a speaker for Dainippon-Sumitomo Pharmaceutical, Mochida Pharmaceutical, and MSD. Kazutaka Shimoda has received research support from Meiji Seika Pharma Co., Pfizer Inc., Dainippon Sumitomo Pharma Co., Ltd., Daiichi Sankyo Co., Otsuka Pharmaceutical Co., Ltd., Astellas Pharma Inc., Novartis Pharma K.K., Eisai Co., Ltd., Takeda Pharmaceutical Co., Ltd. and honoraria from Mitsubishi Tanabe Pharma Corporation, Meiji Seika Pharma Co., Ltd., Dainippon Sumitomo Pharma Co., Ltd., Takeda Pharmaceutical Co., Shionogi & Co., Ltd., Daiichi Sankyo Co., Pfizer Inc. and Eisai Co., Ltd. The companies had no role in the study design, the data collection and analysis, the decision to publish, or the preparation of the manuscript. The remaining authors declare that they have no competing interests to report. ".

i) Please confirm that this does not alter your adherence to all PLOS ONE policies on sharing data and materials, by including the following statement: "This does not alter our adherence to PLOS ONE policies on sharing data and materials.” (as detailed online in our guide for authors http://journals.plos.org/plosone/s/competing-interests). If there are restrictions on sharing of data and/or materials, please state these. Please note that we cannot proceed with consideration of your article until this information has been declared.

Response:

Our competing interests do not affect compliance with PLOS ONE's policies.

The following sentences have been added.

“This does not alter our adherence to PLOS ONE policies on sharing data and materials.”

(Revised Manuscript with Track Changes: Lines 522-523)

Comment:

ii) Please include your updated Competing Interests statement in your cover letter; we will change the online submission form on your behalf.

Response:

The updated Competing Interests statement has been included in the cover letter.

We added the following sentence to the end of the competing interests.

“This does not alter our adherence to PLOS ONE policies on sharing data and materials.”

(Revised Manuscript with Track Changes: Lines 522-523.)

Comment:

Please know it is PLOS ONE policy for corresponding authors to declare, on behalf of all authors, all potential competing interests for the purposes of transparency. PLOS defines a competing interest as anything that interferes with or could reasonably be perceived as interfering with the full and objective presentation, peer review, editorial decision-making, or publication of research or nonresearch articles submitted to one of the journals. Competing interests can be financial or nonfinancial, professional, or personal. Competing interests can arise in relationship to an organization or another person. Please follow this link to our website for more details on competing interests: http://journals.plos.org/plosone/s/competing-interests

Response:

We fully understand the PLOS ONE policy for the declaration of competing interests among corresponding authors.

Comment:

Response:

The ethics committee of Towada City Central Hospital has set restrictions on data sharing.

Please contact the corresponding author for data requests. Upon request, the Ethics Commission will decide whether to share the data.

A contact information for our ethics committee: The institutional review board of the ethics committee of Towada City Hospital (Chairperson of the ethics committee: Dr. Masaru Kudo); Towanda City, Nishi 12-14-8, Aomori Prefecture, Japan, Postal Code 034-0093, Phone +81-716-23-5121, FAX +81-176-23-2999.

(Revised Manuscript with Track Changes: Lines 260-263, 503-510.)

Comment:

a) If there are ethical or legal restrictions on sharing a deidentified data set, please explain them in detail (e.g., data contain potentially identifying or sensitive patient information) and who has imposed them (e.g., an ethics committee). Please also provide contact information for a data access committee, ethics committee, or other institutional body to which data requests may be sent.

Response:

The ethics committee of Towada City Hospital has set restrictions on data sharing because the data contain potentially identifying or sensitive patient information. Please contact the corresponding author for data requests. Upon request, the ethics committee will decide whether to share the data.

A contact information for our ethics committee: The institutional review board of the ethics committee of Towada City Hospital (Chairperson of the ethics committee: Dr. Masaru Kudo); Towanda City, Nishi 12-14-8, Aomori Prefecture, Japan, Postal Code 034-0093, Phone +81-716-23-5121, FAX +81-176-23-2999.

(Revised Manuscript with Track Changes: Lines 260-263, 503-510.)

Comment:

b) If there are no restrictions, please upload the minimal anonymized data set necessary to replicate your study findings as either Supporting Information files or to a stable, public repository and provide us with the relevant URLs, DOIs, or accession numbers. Please see http://www.bmj.com/content/340/bmj.c181.long for guidelines on how to deidentify and prepare clinical data for publication. For a list of acceptable repositories, please see http://journals.plos.org/plosone/s/data-availability#loc-recommended-repositories.

Response:

The ethics committee of Towada City Hospital has set restrictions on data sharing because the data contain potentially identifying or sensitive patient information. Please contact the corresponding author for data requests. Upon request, the ethics committee will decide whether to share the data.

A contact information for our ethics committee: The institutional review board of the ethics committee of Towada City Hospital (Chairperson of the ethics committee: Dr. Masaru Kudo); Towanda City, Nishi 12-14-8, Aomori Prefecture, Japan, Postal Code 034-0093, Phone +81-716-23-5121, FAX +81-176-23-2999.

(Revised Manuscript with Track Changes: Lines 260-263, 503-510.)

Comment:

Response:

We deleted ethical statements other than those listed in the Methods section from our manuscript.

(Revised Manuscript with Track Changes: Lines 488-493, 496-499.)

---

## [Decision Letter · Decision Letter 1]

8 Feb 2021

The combination of MMSE with VSRAD and eZIS has greater accuracy for discriminating mild cognitive impairment from early Alzheimer’s disease than MMSE alone

PONE-D-20-30261R1

Dear Dr. Yasui-Furukori,

We’re pleased to inform you that your manuscript has been judged scientifically suitable for publication and will be formally accepted for publication once it meets all outstanding technical requirements.

Kind regards,

Stephen D. Ginsberg, Ph.D.

Section Editor

PLOS ONE

**Comments to the Author**

1. If the authors have adequately addressed your comments raised in a previous round of review and you feel that this manuscript is now acceptable for publication, you may indicate that here to bypass the “Comments to the Author” section, enter your conflict of interest statement in the “Confidential to Editor” section, and submit your "Accept" recommendation.

Reviewer #1: All comments have been addressed

2. Is the manuscript technically sound, and do the data support the conclusions?

Reviewer #1: Yes

3. Has the statistical analysis been performed appropriately and rigorously? 

Reviewer #1: Yes

4. Have the authors made all data underlying the findings in their manuscript fully available?

Reviewer #1: Yes

5. Is the manuscript presented in an intelligible fashion and written in standard English?

Reviewer #1: Yes

6. Review Comments to the Author

Reviewer #1: All inquiries were successfully addressed. The authors have improved their statistical analysis and addressed the clinical and methodological questions.

7. PLOS authors have the option to publish the peer review history of their article (what does this mean?). If published, this will include your full peer review and any attached files.

Reviewer #1: **Yes: **Dr. Jaime Daniel Mondragon

---

## [Editor Report · Acceptance letter]

11 Feb 2021

PONE-D-20-30261R1 

The combination of MMSE with VSRAD and eZIS has greater accuracy for discriminating mild cognitive impairment from early Alzheimer’s disease than MMSE alone 

Dear Dr. Yasui-Furukori:

I'm pleased to inform you that your manuscript has been deemed suitable for publication in PLOS ONE. Congratulations! Your manuscript is now with our production department. 

Kind regards, 

on behalf of

Dr. Stephen D. Ginsberg 

Section Editor

PLOS ONE